# Essential Workers’ COVID-19 Vaccine Hesitancy, Misinformation, and Informational Needs in the Republic of North Macedonia

**DOI:** 10.3390/vaccines10030348

**Published:** 2022-02-23

**Authors:** Stephen P. Fucaloro, Vahe S. Yacoubian, Rachael Piltch-Loeb, Nigel Walsh Harriman, Tea Burmaz, Metodi Hadji-Janev, Elena Savoia

**Affiliations:** 1Emergency Preparedness Research Evaluation & Practice Program, Harvard T.H. Chan School of Public Health, 90 Smith Street, Boston, MA 02115, USA; stephen.fucaloro@tufts.edu (S.P.F.); vahe.yacoubian@tufts.edu (V.S.Y.); piltch-loeb@hsph.harvard.edu (R.P.-L.); esavoia@hsph.harvard.edu (E.S.); 2Department of Biostatistics, Harvard T.H. Chan School of Public Health, 677 Huntington Avenue, Boston, MA 02115, USA; 3Department of Hygiene and Public Health, Local Health Unit 3 Serenissima, 30100 Venice, Italy; tea.burmaz@aulss3.veneto.it; 4Military Academy, “General Mihailo Apostolski”-Skopje, Ul. “Vasko Karangeleski bb”, University “Goce Delcev Shtip”, 1000 Skopje, North Macedonia; metodi.hadzi-janev@ugd.edu.mk

**Keywords:** COVID-19 vaccine hesitancy, North Macedonia, informational needs, misinformation

## Abstract

The COVID-19 pandemic has resulted in over 5.2 million deaths. Vaccine hesitancy remains a public health challenge, especially in Eastern Europe. Our study used a sample of essential workers living in the Republic of North Macedonia to: (1) Describe rates of vaccine hesitancy and risk perception of COVID-19; (2) Explore predictors of vaccine hesitancy; and (3) Describe the informational needs of hesitant and non-hesitant workers. A phone survey was administered in North Macedonia from 4–16 May 2021. Logistic regression explored associations of COVID-19 vaccine hesitancy with sociodemographic characteristics, non-COVID-19 vaccine hesitancy, previous diagnosis of COVID-19, and individual risk perception of contracting COVID-19. Chi-squared analyses compared differences in informational needs by hesitancy status. Of 1003 individuals, 44% were very likely to get the vaccine, and 56% reported some level of hesitancy. Older age, Albanian ethnicity, increased education, previous COVID-19 diagnosis, acceptance of other vaccines, and increased risk perception of COVID-19 infection were negatively associated with vaccine hesitancy. Results indicated significant differences in top informational needs by hesitancy status. The top informational needs of the hesitant were the freedom to choose to be vaccinated without consequences (57% vs. 42%, *p* < 0.01) and that all main international agencies recommended the vaccine (35% vs. 24%, *p* < 0.01).

## 1. Introduction

COVID-19 is an unprecedented global crisis that has presented unique public health and healthcare challenges to the world. As of 5 December 2021, over 5.2 million people across the globe have died from COVID-19 complications [1]. In December 2020, biotech companies Pfizer and Moderna released vaccines under emergency use authorization, which governments across the globe prioritized for the elderly, the vulnerable, and essential workers [2,3]. However, the value of vaccine protection extends only as far as the public’s willingness to get vaccinated.

As of January 2022, North Macedonia has reported over 8700 deaths attributed to COVID-19 and over 287,000 cases in a population of approximately 2 million people [1]. Despite the plethora of research demonstrating the effectiveness of vaccines in preventing COVID-19 spread and severe clinical outcomes, vaccine hesitancy remains a challenge in the management of this global health crisis [4,5,6]. The North Macedonian Government is running a massive and ongoing nationwide vaccination campaign. However, vaccination rates are lower compared to western nations, potentially because the campaign was not timely and contextually tailored to the informational needs of the population [7]. As of December 2021, official data from the WHO European COVID-19 vaccination monitor reveals that only 38% of the North Macedonian population is fully vaccinated against COVID-19, and this trend is reflected across other nations in the Balkan peninsula such as Bulgaria (25%), Kosovo (42%), Romania (38%), Serbia (45%), Montenegro (37%), and Croatia (48%) [8].

The causes contributing to these low vaccination numbers are complex, and studies focused on understanding COVID-19 vaccination acceptance in Eastern European and Balkan countries are limited. To our knowledge, Slovenia is the only country in which there are extant scientific publications related to COVID-19 vaccine acceptance. Slovenian nursing students showed an acceptance rate of only 33% and a sample of healthcare workers, not including physicians, showed an acceptance rate lower than the general population (50% versus 57% respectively) [9,10]. While examining the pre-COVID vaccine literature, a survey of healthcare workers across six European nations showed Bulgarian healthcare workers reporting the greatest vaccine advocacy, uptake, and overall strong positive sentiments for the flu vaccine, compared to other neighboring nations, in contradiction with the current COVID-19 vaccination rates of approximately 27% [11,12].

The differences in vaccine acceptance patterns across European and Balkan nations are underscored by the sociopolitical intricacies unique to the region. It is presumed that sentiments against vaccinations can be at least partially attributed to an anti-vaccine and anti-government movement that has been growing in this region. These sentiments may have contributed to the European measles outbreak observed in 2018 [13]. Data describing trends in COVID-19 vaccine hesitancy in the Balkans are currently lacking and greatly needed. Specifically, data that describe the socio-demographic predictors of vaccination sentiments and the informational needs of hesitant individuals are needed to establish effective communication strategies that address vaccine beliefs, attitudes, and informational needs that may be related to the spread of misinformation or anti-vaccination sentiments on a per-country basis.

In this study, we focus on the population of the Republic of North Macedonia. To our knowledge, this is the first study aimed at understanding COVID-19 vaccine hesitancy in this nation with specific reference to essential workers’ informational needs. This study aims to: (1) Describe rates of vaccine hesitancy, (2) Explore predictors of vaccine hesitancy, and (3) Describe the information essential workers would need to make them more likely to accept the vaccine.

## 2. Materials and Methods

### 2.1. Study Design

We conducted a survey of essential workers in North Macedonia to assess their informational needs regarding the COVID-19 vaccine. We defined an essential worker as an individual working in any job category within a list of options as presented in Table 1. Categories included the healthcare sector, transportation, food processing, grocery workers, police, firefighters, and volunteers. A local poll company (Rating Agency) was contracted to conduct phone interviews using the Computer Assisted Telephone Interviewing (CATI) technique. The survey was implemented from 4–16 May 2021. At the time of the survey, North Macedonia was experiencing a 7-day average of between 20–30 deaths due to COVID-19. In absence of a roster defining the whole source population of essential workers, non-existent in any country, a representative sample of the adult population was selected, and screening questions were applied to include only respondents that had not taken the vaccine at the time of the survey and were workers in one of the job categories listed in Table 1 (essential workers) in North Macedonia. More specifically, Rating Agency designed a national representative sample by using a multi-stage, random (probability) sample design where a number of sampling points were drawn with the probability proportional to population size (for total coverage of the country) and population density. A multi-staged random sample was constructed by taking a series of simple random samples in stages. The procedure for sample selection was based on the principle of making regional and national representative samples that defined the region in accordance with its definition by the State Statistical Office (NUTS3 the EU16). Namely, according to the geo-demographic structure of the population, the Republic of North Macedonia is divided into eight regions: Skopski, Polog, Pelagonia, Vardarski, Northeast, Southeast, Southwest, and East region, including urban and rural areas. The number of respondents was proportionally allocated according to the total population in each region. All data were collected anonymously, and the study conformed to the guidelines and principles of the Declaration of Helsinki. The study protocol and survey instrument were approved by the Harvard T.H. Chan School of Public Health Institutional Review Board (protocol number IRB20-2032).

### 2.2. Survey Instrument

The survey instrument consisted of 35 questions including the following topics/areas: socio-demographics, COVID-19 and other vaccine hesitancy, risk perception about contracting COVID-19, experience with COVID-19, and informational needs about the vaccine. Questions were translated from English into Macedonian and back-translated into English. A copy of the survey instrument is available upon request to the corresponding author.

### 2.3. Statistical Analysis

We computed descriptive statistics to describe our sample’s socio-demographics and other variables of interest such as COVID-19 and other vaccine hesitancy, risk perception about contracting COVID-19, experience with COVID-19, and informational needs about the vaccine. Predictors of vaccine hesitancy were explored using three logistic regression models. The dependent variable was derived from the answers to the following question: “If you were offered a COVID-19 vaccine within two months from now at no cost to you—how likely are you to take it?” Answers were coded as 1 (hesitant) if the respondent indicated they were: somewhat likely, not interested now but would consider it later on, somewhat unlikely, or very unlikely; and 0 (non-hesitant) if the respondent indicated they were very likely to take it. In model 1, the independent variables included age, sex, and ethnicity. Model 2 included parameters from model 1 with the addition of education as a proxy for socioeconomic status. Model 3 included the parameters from model 2 with the addition of the previous diagnosis of COVID-19, previous non-COVID-19 vaccine hesitancy, and risk perception of contracting COVID-19.

To describe informational needs about the vaccine we analyzed responses to the following question: “What would be important for you to know to make you more likely to take the COVID-19 vaccine?” Respondents were allowed to select the three most important topics to them out of eight possible choices related to vaccine safety and effectiveness, and three choices out of six related to vaccine policies. Chi-Squared tests were used to assess differences in informational needs between hesitant and non-hesitant individuals. The statistical analysis was conducted using SPSS v.28.

## 3. Results

### 3.1. Sample Characteristics

Table 1 displays the descriptive statistics of the sample. We gathered responses from 1003 subjects. Sex was equally distributed with 50% being female and the most represented age group was 35–44 (27%). In terms of representation of ethnic groups, most respondents were Macedonian (77%), followed by Albanian (21%), which is consistent with the distribution of ethnicity in North Macedonia. Ninety-three percent of respondents had a secondary school education or higher. Most respondents (95%) reported being employed and 5% of respondents reported being unemployed or volunteers. The most represented job categories were: public health and healthcare workers (23%), grocery store workers (20%), food processing workers (13%), and teachers/school staff (11%). Responses were obtained from all regions: Skopje (30%), Polog (15%), Pelagonija (11%), Southwest (11%), East (9%), Southeast (9%), Northeast (8%), and Vardar (7%).

### 3.2. COVID-19 and Other Vaccine Hesitancy

When asked the following question: “If you were offered a COVID-19 vaccine within two months from now at no cost to you—how likely are you to take it?” Forty-four percent said they were very likely to take it, 14% somewhat likely, 9.5% were not sure, 4% were somewhat unlikely, 13% were very unlikely, 11% reported they would not take it within two months but would consider it later on, and 4.5% refused to answer. Regarding hesitancy towards other vaccines, 12% of respondents reported that in the past they had refused a vaccine that was recommended to them by a healthcare worker.

### 3.3. Experience of COVID-19 and Risk Perception

Twenty-five percent of respondents said they had been diagnosed with COVID-19 during the pandemic, 55% of respondents had friends or family members who had tested positive and had no or mild symptoms, 35% did not know of anyone who tested positive, 25% had friends or family members who experienced severe symptoms, and 14% knew a friend or family member who had died from COVID-19. Most respondents were concerned about contracting COVID-19 both at work and outside of work (78%).

### 3.4. Predictors of Vaccine Hesitancy

Table 2 presents the results of the logistic regression models. In model 1, sex, ethnicity, and age were significantly associated with vaccine hesitancy. Females had 27% decreased odds of being hesitant compared to males (OR = 0.73, 95% CI: 0.56–0.94). Respondents of Albanian ethnicity had 42% decreased odds of being hesitant compared to individuals who reported Macedonian ethnicity (OR = 0.58, 95% CI: 0.42–0.81). Increased age was associated with decreased odds of vaccine hesitancy; compared to the youngest age group, (18–24) the oldest age group had 77% decreased odds of being hesitant (OR = 0.23, 95% CI: 0.11–0.46). Across all models, age and ethnicity remained significantly associated with vaccine hesitancy, with little change to the magnitude or direction of the associations described above.

Model 2 included the parameters from model 1 with the addition of education level, which was used as a proxy for socioeconomic status. When education was included in the regression model, the male sex was no longer a significant predictor of vaccine hesitancy. Respondents with more than secondary school education (university, master’s programs, and doctorate programs) had 0.32 times the odds of being hesitant compared to individuals that had less than a secondary school education (OR = 0.32, 95% CI: 0.18–0.58).

Model 3 included parameters from both model 1 and model 2 with the addition of previous non-COVID-19 vaccine hesitancy, previous diagnosis of COVID-19 infection, and degree of concern about contracting COVID-19. In this model, educational attainment remained significantly negatively associated with vaccine hesitancy. Those with prior vaccine hesitancy had lower odds of being hesitant to take the COVID-19 vaccine. Respondents reporting that they had not avoided a vaccine recommended to them in the past had 0.51 times the odds of being hesitant compared to individuals who were hesitant (OR = 0.51, 95% CI: 0.36–0.8). Individuals who reported they had been diagnosed with COVID-19 in the past had 32% decreased odds of being hesitant compared to individuals who did not receive a diagnosis (OR = 0.68, 95% CI: 0.5–0.93). Respondents reporting concerns about contracting COVID-19 at either their workplace or home had 0.41 times the odds of being hesitant compared to respondents that were not concerned (OR = 0.41, 95% CI: 0.21–0.79). Furthermore, respondents reporting concerns about contracting COVID-19 at both their workplace and their homes had 76% decreased odds of being hesitant to vaccination compared to non-concerned individuals (OR = 0.24, 95% CI: 0.14–0.4).

### 3.5. Informational Needs

Respondents were asked what would be important for them to know to make them more likely to receive the vaccine. Each respondent was asked to choose the three most important topics that were unordered and mutually exclusive. Table 3 displays the three most frequently selected topics by hesitancy status, and Figure 1 and Figure 2 illustrate the differences between hesitant and non-hesitant individuals.

Among the hesitant, the three most frequently selected topics—related to the safety and effectiveness of the vaccine—for which respondents wanted more information were: (1) The FDA, CDC, and WHO recommend and agree it is safe (35%), (2) the vaccine works in stopping the spread of COVID-19 from one person to another (35%), and (3) The vaccine cannot cause any immediate or long-term harm (33%). Among the non-hesitant respondents, the three most frequently selected topics in this category were (1) My risk of getting sick with COVID-19 is bigger than the risk of side effects from the vaccine (40%), (2) The vaccine works in protecting me from COVID-19 (37%), and 3) The vaccine works in stopping the spread of COVID-19 from one person to another (35%).

Respondents were also asked to select topics for which they wanted reassurance related to the vaccine policies. The most frequently selected topics among hesitant respondents were (1) I will be free to choose if I get the vaccine or not with no consequences (57%), (2) Once vaccinated, I will be able to live my life with no restrictions (37%), and (3) Everybody will have equal access to the vaccine regardless of income, race, or insurance status (32%). Interestingly, the non-hesitant shared these same concerns in their top three, however, the order was different: (1) Once vaccinated I will be able to live my life with no restrictions (62%), (2) Everybody will have equal access to the vaccine regardless of income, race, or insurance status (44%), and (3) I will be free to choose if I get the vaccine or not with no consequences (42%).

We also noted which of these top three informational needs had significantly different percentages by hesitancy status. With regard to the vaccine safety-related informational needs of the hesitant, these individuals, compared to the non-hesitant, had a higher percentage of respondents who indicated they wanted to have some reassurance that main international agencies (i.e., FDA, WHO, EMA) were recommending the vaccine and that they would be free to choose to get the vaccine or not without consequences (35% vs. 24%; *p* < 0.01).

Regarding the vaccine safety-related informational needs of the non-hesitant, these individuals, compared to the hesitant, had a higher percentage of respondents indicating they needed to be reassured that getting sick with COVID-19 is worse than the risk of side effects from the vaccine (40% vs. 29%; *p* < 0.01) and that the vaccine will protect them from COVID-19 (37% vs. 29%; *p* < 0.05).

While the hesitant and non-hesitant shared the same top three vaccine policy-related informational needs, there were significant differences in their percentages. The hesitant, compared to the non-hesitant, had a higher percentage of respondents who indicated they wanted to be free to choose if they received the vaccine or not with no consequences (57% vs. 42%, *p* < 0.01). The non-hesitant had a higher percentage of respondents who indicated they wanted to be reassured that they would live with no restrictions once vaccinated (62% vs. 37% *p* < 0.01) and that everyone would have equal access to the vaccine (44% vs. 32%; *p* < 0.01).

When comparing informational needs related to vaccine misinformation between hesitant and non-hesitant respondents, there was no statistically significant difference in the frequency of selection. The response, “There is no other reason why we have so many people sick (i.e., 5G technology or other factors)” was selected by 29% of hesitant individuals and 25% of non-hesitant individuals. The response, “It is impossible to get COVID-19 or any other disease from the vaccine itself or its components” was selected by 21% of hesitant respondents and 18% of non-hesitant respondents.

## 4. Discussion

As of January 2022, approximately 40% of adults in the Republic of North Macedonia have been fully vaccinated against COVID-19. The rate is well short of other Western European countries - most of which have vaccination rates over 60%. The underlying causes contributing to the lack of vaccination coverage in North Macedonia are complex. North Macedonia has struggled with vaccine supply shortages since the early days of its vaccine distribution campaign in mid-February. In addition, limited local capacity in launching timely and tailored awareness campaigns may have affected certain socially disadvantaged groups, such as the Roma population, contributing to a lack of coverage. Overall, general vaccine hesitancy remains one of the largest barriers to vaccine uptake in N. Macedonia [14,15].

Our survey was conducted in May 2021 when approximately 4% of the country’s 2.1 million residents had been vaccinated, this rate is now close to 40%. As such, the survey was implemented at a time of high demand and low supply, yet our results indicate that vaccine hesitancy may have impacted vaccine uptake at the time of the survey and in subsequent months. Our survey found approximately 56% of respondents reported some degree of hesitancy in getting the vaccine, which is consistent with a similar survey conducted in the United States at the start of the vaccine distribution campaign [13]. When we explored the predictors of such hesitancy, previous vaccine acceptance, having been diagnosed with COVID-19 in the past, and being concerned with contracting the disease were negatively associated with hesitancy. Both hesitant and non-hesitant individuals reported that they needed more information about the vaccine, however, they had different priorities in terms of what it was important for them to know. The hesitant needed reassurances that the major public health institutions of the world agreed that the vaccine was safe and that they would be free to choose to get the vaccine or not, while those who were more amenable to vaccination wanted to have more information about the risks and benefits of the vaccine and reassurances that once vaccinated, they could live a life with no restrictions.

We recognize that this study has limitations. First, our sample is not representative of all essential workers in the Republic of North Macedonia. However, a representative sample of essential workers would be very difficult to obtain given the multitude of job categories included in the list of essential workers and the lack of data on how many people work within each category in the nation. Our results should be interpreted taking into consideration the time when the survey was implemented—vaccine hesitancy, as well as its determinants, may have changed since May 2021. The lack of longitudinal data does not allow us to study changes in the willingness to be vaccinated, therefore we do not know if those who were hesitant in May are the same people who are hesitant now, or if the informational needs we identified are reflective of current concerns. While our survey did capture risk perception about COVID-19, we did not include questions regarding awareness of COVID-19 complications, which may impact vaccine hesitancy—future research should aim to address this relationship. Finally, our results may not be comparable to other national polls or surveys because of potential differences in the survey methods, sample populations, and questions related to vaccination intent.

Despite its limitations, we believe this study provides useful information to further understand vaccine hesitancy in the Balkans and how to enhance communication strategies that address misinformation. As Balkan governments have scrambled to secure COVID-19 vaccines, this region has become a breeding ground for anti-vaccine movements. The Balkans has long been a hotbed of misinformation, fueled by low levels of trust in government and other institutions [16]. Regarding potential exposure to misinformation, our survey indicates that about 1 out of 5 respondents believed they could get COVID-19 from the vaccine itself and that COVID-19 could be caused by 5G. Interestingly, these results are also consistent with what was reported in a similar study conducted in the United States [17]. Beliefs in unproven theories may be the result of a lack of trust in institutions, due to the politicized response to the pandemic or anti-COVID-19-accompanied corruption scandals [16,18,19,20]. The hampered EU integration has led to skepticism towards Western countries distributing the vaccine; the vaccines and COVID-19 restrictive measures have been spreading as a Western-based conspiracy to impose control over the masses fueling anti-vax movements in a context where the local government has limited capacity to counter misinformation [21,22]. Understanding reasons for hesitancy and the spread of misinformation using population surveys—similar to the one presented in this study—as well as monitoring social media is important to enhance the government’s capability to effectively communicate to the public.

## Figures and Tables

**Figure 1 vaccines-10-00348-f001:**
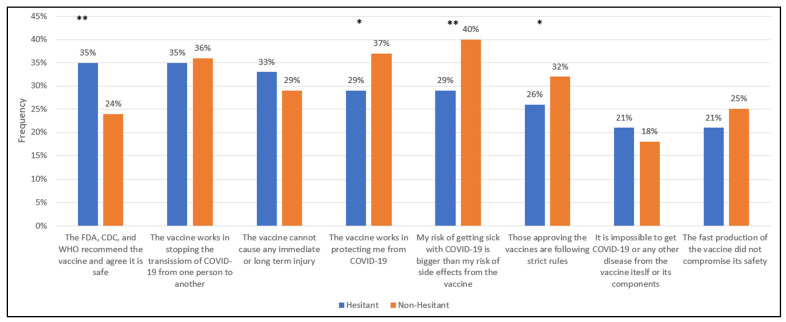
Informational needs of the respondents ranked in descending order for hesitancy: topics related to vaccine safety and efficacy. * Indicates χ^2^ test between hesitant and non-hesitant proportions is significant at the 0.05 alpha level. ** Indicates χ^2^ test between hesitant and non-hesitant proportions is significant at the 0.01 alpha level.

**Figure 2 vaccines-10-00348-f002:**
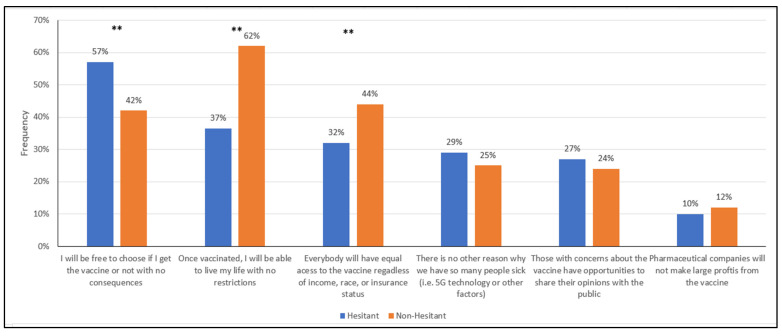
Informational needs of the respondents ranked in descending order for hesitancy: topics related to vaccination policies. * Indicates χ^2^ test between hesitant and non-hesitant proportions is significant at the 0.05 alpha level. ** Indicates χ^2^ test between hesitant and non-hesitant proportions is significant at the 0.01 alpha level.

**Table 1 vaccines-10-00348-t001:** Sample characteristics and vaccine hesitancy (N = 1003).

Variable		*n* (%)
Vaccine Hesitancy		
	I would not take it within 2 months, but maybe later on	106 (11)
	Very unlikely	133 (13)
	Somewhat unlikely	40 (4)
	I am not sure	95 (9.5)
	Somewhat likely	144 (14)
	Very Likely	439 (44)
**Demographics**		
Age		
	18–24	56 (5.6)
	25–34	202 (20)
	35–44	275 (27)
	45–54	266 (27)
	55+	204 (20)
Sex		
	Male	496 (50)
	Female	507 (50)
Ethnicity		
	Macedonian	776 (77)
	Albanian	207 (21)
	Serb	7 (0.7)
	Turkish	5 (0.5)
	Vlach	3 (0.3)
	Roma	2 (0.2)
	Bosniak	2 (0.2)
	Another	1 (0.1)
Region		
	Vardar	67 (6.7)
	East	91 (9.1)
	Southwest	107 (11)
	Southeast	87 (8.7)
	Pelagonija	110 (11)
	Polog	152 (15)
	Northeast	85 (8.5)
	Skopje	304 (30)
**Socioeconomic Status**		
Education		
	No education	1 (0.1)
	Primary	32 (3.2)
	Three-year Secondary	32 (3.2)
	Secondary	591 (59)
	Higher Education	48 (4.8)
	University, Master or PhD	295 (29)
Occupation		
	Unemployed	6 (0.6)
	Hospital and emergency department workers	54 (5.4)
	Nursing home, long-term care, and home health care workers	16 (1.6)
	Public health workers	79 (7.9)
	Emergency Medical Services workers	19 (1.9)
	Prisons workers	4 (0.4)
	Sanitation workers	15 (1.5)
	Vaccine manufacturing workers	1 (0.1)
	Other health care workers	17 (1.7)
	Pharmacy workers	48 (4.8)
	Teachers and school staff (including childcare and K-12)	110 (11)
	Food processing workers	125 (13)
	Grocery store workers	205 (20)
	Postal and shipping workers	76 (7.6)
	Public transportation workers	40 (4)
	Private transportation workers	70 (7)
	Police or firefighters	63 (6.3)
	Other first responders	8 (0.8)
	Volunteer (i.e., CERT, MRC, Red Cross, etc.)	47 (4.7)
**Experience with COVID-19**		
Past Refusal of Non-COVID-19 vaccines		
	Yes	117 (12)
	No	770 (77)
	I don’t remember	78 (7.8)
	I don’t Know	26 (2.6)
Previous COVID-19 diagnosis		
	Yes	247 (25)
	No	749 (75)
	I don’t Know	7 (0.7)
Friends and family experience with COVID-19		
	Friend or family member tested positive with no or mild symptoms	554 (55)
	Friend or family member tested positive and had severe symptoms	252 (25)
	Friend or family member died from COVID-19	138 (14)
	Friend or family member lost their job/had a salary cut due to COVID-19	122 (12)
	Friends or family experienced none of the above	346 (35)
	I don’t know	15 (1.5)
	Refuse to Answer	2 (0.2)
Level of Concern for contracting COVID-19		
	At work or outside work	93 (9.3)
	Both at work and outside work	783 (78)
	No concern	127 (13)

**Table 2 vaccines-10-00348-t002:** Multivariable models of Vaccine Hesitancy Predictors.

Variables	Model 1		Model 2		Model 3	
**Age**	**OR**	**95% C.I.**	**OR**	**95% C.I.**	**OR**	**95% C.I.**
18–24	*Ref*		*Ref*		*Ref*	
25–34	0.36 *	(0.18–0.73)	0.47 *	(0.23–0.95)	0.46 *	(0.22–0.94)
35–44	0.27 **	(0.14–0.54)	0.34 *	(0.17–0.68)	0.34 *	(0.17–0.68)
45–54	0.28 **	(0.14–0.56)	0.34 *	(0.17–0.68)	0.34 *	(0.17–0.7)
55+	0.23 **	(0.11–0.46)	0.28 **	(0.14–0.57)	0.28 **	(0.13–0.57)
**Sex**						
Male	*Ref*		*Ref*		*Ref*	
Female	0.73 *	(0.56–0.94)	0.79	(0.6–1.03)	0.85	(0.65–1.12)
**Ethnicity**						
Macedonian	*Ref*		*Ref*		*Ref*	
Albanian	0.58 *	(0.42–0.81)	0.56 **	(0.4–0.79)	0.5 **	(0.35–0.72)
Other	0.81	(0.32–2.08)	0.77	(0.3–1.99)	0.74	(0.28–1.97)
**Education**						
Less than secondary school			*Ref*		*Ref*	
Secondary school			0.62	(0.35–1.1)	0.72	(0.4–1.3)
More than secondary school			0.32 **	(0.18–0.58)	0.4 *	(0.22–0.73)
**Other Vaccine Hesitancy**						
Previous non-COVID-19 Vaccine Hesitancy					*Ref*	
Previous non-COVID-19 Vaccine Acceptance					0.51 *	(0.36–0.8)
I don’t know/Don’t remember/refuse to answer					0.65	(0.36–1.17)
**Previous COVID-19 Diagnosis**						
Never diagnosed with COVID-19					*Ref*	
Diagnosed with COVID-19 in the past					0.68 *	(0.5–0.93)
**COVID-19-Related Concern**						
Not concerned with contracting COVID-19					*Ref*	
Concerned with contracting COVID-19 at work or outside work					0.41 *	(0.21–0.79)
Concerned with contracting COVID-19 at work and outside work					0.24 **	(0.14–0.4)

* Indicates that the Wald test statistic comparing this estimate to the reference group is significant at the 0.05 alpha level. ** Indicates that the Wald test statistic comparing this estimate to the reference group is significant at the 0.01 alpha level.

**Table 3 vaccines-10-00348-t003:** Informational needs related to vaccine safety, effectiveness, and policies by vaccine hesitancy status (*n* = 957).

Informational Needs Related to Vaccine Safety and Effectiveness	
Hesitant individuals (*n* = 518)	The FDA, CDC, and WHO recommend and agree it is safe (35%) ** The vaccine works in stopping the spread of COVID-19 from one person to another (35%) The vaccine cannot cause any immediate or long-term harm (33%)
Non-Hesitant individuals (*n* = 439)	My risk of getting sick with COVID-19 is bigger than the risk of side effects from the vaccine (40%) ** The vaccine works in protecting me from COVID-19 (37%) * The vaccine works in stopping the spread of COVID-19 from one person to another (35%)
**Informational Needs Related to Vaccine Policies**	
Hesitant individuals (*n* = 518)	(1) I will be free to choose if I get the vaccine or not with no consequences (57%) ** (2) Once vaccinated I will be able to live my life with no restrictions (37%) ** (3) Everybody will have equal access to the vaccine regardless of income, race, or insurance status (32%) **
Non-hesitant individuals (*n* = 439)	(1) Once vaccinated I will be able to live my life with no restrictions (62%) ** (2) Everybody will have equal access to the vaccine regardless of income, race, or insurance status (44%) ** (3) I will be free to choose if I get the vaccine or not with no consequences (42%) **

* Indicates χ^2^ test between hesitant and non-hesitant proportions is significant at the 0.05 alpha level. ** Indicates χ^2^ test between hesitant and non-hesitant proportions is significant at the 0.01 alpha level.

## Data Availability

The data presented in this study are available on request from the corresponding author.

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
