# Peer review of "Essential Workers’ COVID-19 Vaccine Hesitancy, Misinformation, and Informational Needs in the Republic of North Macedonia"

_vaccines, 2022, doi:10.3390/vaccines10030348_

Round 1
Reviewer 1 Report
Fucaloro and colleagues studied the COVID-19 vaccine hesitancy in the Balkans. The authors conducted a survey of essential workers in North Macedonia which consisted of 35 questions including the following topics/ areas: socio-demographics, COVID-19 and other vaccine hesitancy, risk perception about contracting COVID-19, experience with COVID-19, and informational needs about the vaccine. The author found that 56% of respondents reported some degree of hesitancy in getting the vaccine. Hesitant and non-hesitant individuals require more information about the vaccine.
The manuscript could be beneficial especially for people who living in the Balkans. I have some questions
1) My major concerns in Table 2
a) The statistic is table 2 is not clear. For examples, below the table the authors mentioned that * p<0.05; p<0.01. Inside the table there are stars put either * or** , but non of the values matched with the data below the table.
b) Can be the awarness about COVID19 complications one of the predicator factor for vaccine hesitancy?
c) WHat is the mortality rate associated with COVID19 among family members/ friends for prople taking this survey?
2) In introduction: please add information about infection rate, complications, mortality rate assocaited with COVID19 infection in the Balkans.
Author Response
Fucaloro and colleagues studied the COVID-19 vaccine hesitancy in the Balkans. The authors conducted a survey of essential workers in North Macedonia which consisted of 35 questions including the following topics/ areas: socio-demographics, COVID-19 and other vaccine hesitancy, risk perception about contracting COVID-19, experience with COVID-19, and informational needs about the vaccine. The author found that 56% of respondents reported some degree of hesitancy in getting the vaccine. Hesitant and non-hesitant individuals require more information about the vaccine.
The manuscript could be beneficial especially for people who living in the Balkans. I have some questions
1) My major concerns in Table 2
- a) The statistic is table 2 is not clear. For examples, below the table the authors mentioned that * p<0.05;p<0.01. Inside the table there are stars put either *or**, but none of the values matched with the data below the table.
We sincerely appreciate the reviewer’s feedback on table 2. These stars refer to Wald test statistics comparing the parameter estimate for the category to the reference group. We have updated the key to reflect this.
We have also double checked the results below the table – all the OR and 95% CI match the table. If the changes to table 2 are not sufficient, we would appreciate further feedback on this point.
- b) Can be the awareness about COVID19 complications one of the predictor factors for vaccine hesitancy?
This is an important point, and we agree with the reviewer that this relationship requires investigation. While we did include questions about risk perception of COVID-19 in our model, our survey did not have questions measuring awareness of COVID-19 complications that would allow us to investigate this relationship. We have added to the limitations section to address this (Lines 375-378)
- c) What is the mortality rate associated with COVID19 among family members/ friends for people taking this survey?
Thank you for this helpful feedback – we have updated the results section to include the percentage of respondents who indicated they knew a friend or family member who had died from COVID-19 (lines 174-175).
2) In introduction: please add information about infection rate, complications, mortality rate associated with COVID19 infection in the Balkans.
We have now included data specific to N. Macedonia. Given the study was conducted in this country we believe it is more appropriate to refer to country-specific data rather than the overall Balkan region (please see lines 41-42, and 91-93).
Reviewer 2 Report
This study explores the COVID-19 vaccine hesitancy in North Macedonia. The manuscript is well written. However, here are few concerns to address:
- This study was performed in May 2021 which means the situation might have already changed and may not represent the current status. This needs to be acknowledged.
- It states, the study was carried out among essential workers, but do not provide their details. Who are they, how the participants were selected/listed, how the study was randomized and what is the total population of essential workers?
- Since the figures 1 and 2 show the data included in table 3, table 3 may be removed.
Author Response
This study explores the COVID-19 vaccine hesitancy in North Macedonia. The manuscript is well written. However, here are few concerns to address:
- This study was performed in May 2021 which means the situation might have already changed and may not represent the current status. This needs to be acknowledged.
Thank you for this important suggestion – we have now acknowledged in the limitations portion of the discussion. See lines 348 and 370-372.
- It states, the study was carried out among essential workers, but do not provide their details. Who are they, how the participants were selected/listed, how the study was randomized and what is the total population of essential workers?
We agree with the reviewer that this would help the manuscript. We have now provided more details on the definition of essential worker, job categories (lines 85-88) and on the sampling procedure (lines 97-108).
- Since the figures 1 and 2 show the data included in table 3, table 3 may be removed.
We agree with the reviewer that the information provided in table 3 is redundant – however with this paper we aim to reach practitioners as well and not only individuals with a research background, practitioners which may not have the knowledge on how to interpret a graph and may find a simple table with the list of identified issues easier to read. Unless the reviewer strongly opposes to table 3 we would like to keep it.
Round 2
Reviewer 1 Report
The revised manuscript includes my suggestions.